# Proteomic Landscape and Deduced Functions of the Cardiac 14-3-3 Protein Interactome

**DOI:** 10.3390/cells11213496

**Published:** 2022-11-04

**Authors:** Jia-Hua Qu, Kirill V. Tarasov, Khalid Chakir, Yelena S. Tarasova, Daniel R. Riordon, Edward G. Lakatta

**Affiliations:** Laboratory of Cardiovascular Science, Intramural Research Program, National Institute on Aging, National Institutes of Health, Baltimore, MD 21224, USA

**Keywords:** 14-3-3, interactome, protein–protein interaction, mitochondria, metabolism, protein quality control, homeostasis, left ventricle, network

## Abstract

*Rationale*: The 14-3-3 protein family is known to interact with many proteins in non-cardiac cell types to regulate multiple signaling pathways, particularly those relating to energy and protein homeostasis; and the 14-3-3 network is a therapeutic target of critical metabolic and proteostatic signaling in cancer and neurological diseases. Although the heart is critically sensitive to nutrient and energy alterations, and multiple signaling pathways coordinate to maintain the cardiac cell homeostasis, neither the structure of cardiac 14-3-3 protein interactome, nor potential functional roles of 14-3-3 protein–protein interactions (PPIs) in heart has been explored. *Objective*: To establish the comprehensive landscape and characterize the functional role of cardiac 14-3-3 PPIs. *Methods and Results*: We evaluated both RNA expression and protein abundance of 14-3-3 isoforms in mouse heart, followed by co-immunoprecipitation of 14-3-3 proteins and mass spectrometry in left ventricle. We identified 52 proteins comprising the cardiac 14-3-3 interactome. Multiple bioinformatic analyses indicated that more than half of the proteins bound to 14-3-3 are related to mitochondria; and the deduced functions of the mitochondrial 14-3-3 network are to regulate cardiac ATP production via interactions with mitochondrial inner membrane proteins, especially those in mitochondrial complex I. Binding to ribosomal proteins, 14-3-3 proteins likely coordinate protein synthesis and protein quality control. Localizations of 14-3-3 proteins to mitochondria and ribosome were validated via immunofluorescence assays. The deduced function of cardiac 14-3-3 PPIs is to regulate cardiac metabolic homeostasis and proteostasis. *Conclusions*: Thus, the cardiac 14-3-3 interactome may be a potential therapeutic target in cardiovascular metabolic and proteostatic disease states, as it already is in cancer therapy.

## 1. Novelty and Significance

### 1.1. What Is Known?

The 14-3-3 proteins interact with many proteins in non-cardiac cell types to regulate multiple signaling pathways, particularly those relating to energy and protein homeostasis.

The 14-3-3 network is a therapeutic target of critical metabolic and proteostatic signaling in cancer and neurological diseases.

The heart is critically sensitive to nutrient and energy alterations, and multiple signaling pathways coordinate to maintain the cardiac cell homeostasis.

### 1.2. What New Information Does This Article Contribute?

In mouse heart, a comprehensive 14-3-3 protein–protein interactions (PPIs) network was constructed, which can be used in future research.

In the heart, 14-3-3 proteins localize within mitochondria to a large degree and play an important role in the regulation of mitochondrial functions and metabolism.

In the heart, 14-3-3 proteins interact with protein synthesis related proteins, e.g., ribosome proteins, as well as protein degradation related proteins, e.g., chaperone binding proteins, thus involved in the regulation of protein quality control.

An integrated signaling pathway and metabolic process network, generated in mouse heart, emphasizes the potential importance of 14-3-3 PPIs in the maintenance of energy homeostasis and cardiac homeostasis.

Our findings create a Segway to future cardiac research on the regulation of 14-3-3 interaction network as the therapeutic target.

We dissected the structure of cardiac 14-3-3 PPIs, deducing their functions to have a major role to regulate cardiac metabolic homeostasis and proteostasis. Our findings proposed the regulation of the 14-3-3 interaction network as a therapeutic target in metabolic and proteostatic aspects of cardiovascular disease states, as it already is in cancer.

## 2. Introduction

14-3-3 proteins, first discovered in bovine brain homogenates, were named on the basis of their elution in 14th fraction of DEAE-cellulose chromatography and their migration to positions 3.3 of the follow-up electrophoresis [1]. Because the first recognized function of this group of proteins was to activate tyrosine and tryptophan hydroxylases, two important enzymes in the biosynthesis of neurotransmitters [2], the 14-3-3 gene was also called Tyrosine 3-Monooxygenase/Tryptophan 5-Monooxygenase Activation Protein (Ywha*).

The 14-3-3 protein family consists of seven members: β, ε, γ, η, τ/θ, ζ, and σ (the isoform α or δ is the phosphorylated form of β or γ respectively) in non-cardiac cells and is highly conserved in eukaryotes [3,4]. Different 14-3-3 isoforms display distinct subcellular localizations, most existing in the cytoplasm, although some can enter the nucleus, and others have been reported in a few researches to co-localize with mitochondrial components [5].

14-3-3 proteins are expressed in many tissues, including brain, liver, and testes [6]. and are involved in metabolic processes that maintain the whole body energy and nutrient homeostasis [7]. Mass spectrometry (MS) has been used to elucidate 14-3-3 interactions in various contexts [8,9,10,11,12,13], including many oral cancer cell-lines [8] and Tet-inducible PCF cells overexpressing V5-3C-MYC tagged 14-3-3I and non-tagged 14-3-3II [9] with 14-3-3 protein pull-down as bait. Alternatively, 14-3-3 proteins have been also discovered to interact with proteins in MS data when these proteins were used as bait [10,11,12,13]. In non-cardiac cells, multifunctional 14-3-3 protein–protein interactions (PPIs) transduce signals that affect signaling within multiple pathways by sensing and reacting to stimuli that are evoked by environmental stress and impact energy and nutrients [14,15,16,17,18,19,20], and regulating the cellular compartment trafficking, stability and activity of their target proteins [4,21]. Because mutations or dysfunctions of 14-3-3 proteins or abnormalities of the 14-3-3 PPIs contribute to many pathologies and disorders [22,23], 14-3-3 proteins are regarded as the therapeutic targets in many non-cardiac diseases, most of which are metabolic or age-related diseases, e.g., Parkinson diseases, Alzheimer disease and cancers [22,23,24].

The heart not only pumps blood, but also serves as a checkpoint for a complex signaling network in the body [25,26], and the crosstalk between heart with the nervous and endocrine systems maintains the body’s inner homeostasis [27,28]. Although different 14-3-3 isoforms have been identified in human heart tissue [22], and the mutant of one isoform, 14-3-3h, is reported to exacerbate diabetic cardiomyopathy in mouse [29,30,31,32], the comprehensive proteome landscape of the cardiac 14-3-3 interactome, or the deduced functions of the cardiac 14-3-3 network have not been elaborated. Other studies that address 14-3-3 binding in the heart [33,34,35] had focused on a specific binding client. Heart cells, like other non-cardiac cell types throughout nature, are particularly sensitive to energy and nutrient alterations [36,37,38], so we hypothesized that the heart is also likely to be influenced by numerous 14-3-3 PPIs. The present study aimed to discover numerous potential 14-3-3 PPIs by combining co-immunoprecipitation (co-IP) and MS with co-immunolabeling to discover which heart proteins interact with 14-3-3 proteins to establish the global 14-3-3 PPIs landscape in mouse heart left ventricle (LV). Making full use of multiple bioinformatic analysis tools (details in Figure 1 and Section 3), we comprehensively deduced the major functional pathways in mouse heart that are modulated by cardiac 14-3-3 PPIs.

## 3. Methods

### 3.1. Analysis of Public Transcriptomic Data of Mouse Heart

The transcriptomic data of three-month old wildtype C57BL/6J male mouse hearts [39] (BioProject: PRJNA264807; SRA: SRP049245; Series: GSE62689; Samples: GSM1531478 (unstressed_ntg_1), GSM1531479 (unstressed_ntg_2), GSM1531480 (unstressed_ntg_3), GSM1531481 (unstressed_ntg_4)) were downloaded from NCBI GEO database. The TPM values, representing the normalized RNA expression level, were calculated in Partek^®^ Flow^®^ software, version 9.0 Copyright©; 2020 Partek Inc., St. Louis, MO, USA, by utilizing the computational resources of the NIH HPC Biowulf cluster (http://hpc.nih.gov, accessed on 15 October 2021). The organized data were in Appendix A. The TPM values of 14-3-3 isoforms (*Ywhab*, *Ywhae*, *Ywhag*, *Ywhah*, *Ywhaq* and *Ywhaz*) were compared and displayed in Figure 1A. The transformed mean TPM, log2(meanTPM + 1), was used in circos and correlation scatter plots.

### 3.2. Protein Expression Level Analysis of 14-3-3 Isoforms in Mouse

The proteomic data of 3-month old wildtype C57BL/6J male mouse hearts were downloaded from the Appendix A in a SILAC proteome project [40]. The organized data were in Appendix A. The endogenous light (L) intensities of heart protein of 14-3-3β, 14-3-3ε, 14-3-3γ, 14-3-3η, 14-3-3θ, and 14-3-3ζ were compared and displayed in Figure 1B. The transformed mean L intensities of heart protein, log2(meanIntensity + 1), were used in circos and correlation scatter plots.

### 3.3. Animals

All studies were performed on male 3-month old C57/BL6 mice. All studies were conducted in accordance with the Guide for the Care and Use of Laboratory Animals published by the National Institutes of Health (NIH Publication no. 85-23, revised 1996). The experimental protocols were approved by the Animal Care and Use Committee of the National Institutes of Health (protocol #457-LCS-2019).

### 3.4. Extraction of Heart Tissue and Co-Immunoprecipitation (Co-IP) with Pan-14-3-3 Antibody (Ab) and IgG Ab

Twelve three-month old male C57 mice were euthanized with pentobarbital and the hearts were extracted. The left ventricle tissues were isolated, washed in ice cold PBS then snap frozen in liquid nitrogen, and stored in tubes at −80 °C.

LV tissues were removed from −80 °C and kept on dry ice. 40 μL of Pierce Lysis buffer (+Protease Inhibitor) was added for every mg of tissue. The tissue was homogenized using a polytron with disposable tips (10–20 s). Homogenized samples were incubated in lysis buffer for an additional 30 min at 4 °C. The lysate was then transferred to 5 mL tubes and cellular debris was pelleted by centrifugation (14,000× *g* for 15 min at 4 °C). Supernatants were collected into new 15 mL conical tubes.

Total protein was quantified by BCA, and samples were pooled randomly into 4 samples, with approximately 21 mg of total protein per pool (Appendix A). For pre-clearing, 50 μL of Protein G resin (pre-washed with PBS) (Genscript #L00209) was added to each sample and was incubated for 15 min at 4 °C. The resin was pelleted by centrifugation for 5 min at 2000 rpm. The supernatants were transferred to new conical tubes, where 16 μg of the pan-14-3-3 Ab (SCBT sc-133233 Monoclonal Antibody) or 16 μg isotype IgG control antibody (R&D Systems MAB002) was added. The Ab was incubated for 1 h at 4 °C with gentle rocking. After that, 40 μL of pre-washed Protein G resin was added, and the samples were incubated for an additional hour together with the resin. Resin was pelleted by centrifugation at 800× *g* for 5 min. The supernatant was removed, and the resin was washed four times with 200 μL of lysis buffer. After the final wash, the beads were resuspended in 50 μL of 1X SDS loading buffer.

### 3.5. SDS-PAGE Coomassie Staining

IP resin in loading buffer was heated at 100 °C for 15 min and 50% separated ~1.5 cm on a 10% Bis-Tris Novex mini-gel (Invitrogen, Waltham, MA, USA) using the MES buffer system. The gel was stained with Coomassie for protein visualization and quantification. The gel image demonstrated the same amount of proteins in the samples (Appendix AA).

### 3.6. Mass Spectrometry and Data Processing

Each lane of the SDS-PAGE gel was excised into ten equally sized segments. Gel pieces were processed using a robot (ProGest, DigiLab) with the following protocol: (1) Washed with 25 mM ammonium bicarbonate followed by acetonitrile; (2) Reduced with 10 mM dithiothreitol at 60 °C followed by alkylation with 50 mM iodoacetamide at RT; (3) Digested with trypsin (Promega, Madison, WI, USA) at 37 °C for 4 h; (4) Quenched with formic acid and the supernatant was analyzed directly without further processing.

Each gel digest was analyzed by nano LC/MS/MS with a Waters NanoAcquity HPLC system interfaced to a ThermoFisher Q Exactive. Peptides were loaded on a trapping column and eluted over a 75 μm analytical column at 350 nL/min; both columns were packed with Luna C18 resin (Phenomenex, Torrance, CA, USA). A 30 min gradient was employed (5 h total per sample). The mass spectrometer was operated in data-dependent mode, with MS and MS/MS performed in the Orbitrap at 70,000 FWHM and 17,500 FWHM resolution, respectively. The fifteen most abundant ions were selected for MS/MS.

Data were searched using a local copy of Mascot with the following parameters: (1) Enzyme: Trypsin; (2) Database: Swissprot Mouse (concatenated forward and reverse plus common contaminants); (3) Fixed modification: Carbamidomethyl (C); (4) Variable modifications: Oxidation (M), Acetyl (Protein N-term), Deamidation (NQ), Pyro-Glu (N-term Q); (5) Mass values: Monoisotopic; (6) Peptide Mass Tolerance: 10 ppm; (7) Fragment Mass Tolerance: 0.02 Da; (8) Max Missed Cleavages: 2. Mascot DAT files were parsed into the Scaffold software for validation, filtering and to create a nonredundant list per sample. Data were filtered at 1% protein and peptide level FDR and requiring at least two unique peptides per protein. At last, a total of 912 mouse proteins were detected with two or more unique peptides at the protein-level false discovery rates indicated above (based on forward/decoy database searching) (Appendix A).

To compare across samples and proteins, we calculated the normalized counts (NSAF) in two steps: (1) calculating SAF by dividing SpC to the corresponding protein length, and (2) calculating NSAF by dividing SAF to the summed SAF values of the corresponding sample. After that, we transformed the NSAF into log2NSAF to make the data’s distribution closer to the nominal distribution. The information about the 912 proteins is in Appendix A. Both of the NSAF (Appendix AB) and log2NSAF (Appendix A) were used to display the data distribution. Furthermore, the SpC was also normalized to another form of normalized count in R package, countdata [41,42] (Appendix AD). The normalized countdata were used in sample distance and PCA analyses (Appendix AE,F).

### 3.7. Validation of Binding to 14-3-3 via Co-IP and Western Blotting

Co-IP and Western blotting experiments were conducted in *two batches*. The detailed steps of co-IP were described in Method 4. The major differences between the two batches were: (1) a larger proportion of lysate was incubated with the antibody to 14-3-3 in batch 1 than that in batch 2; (2) the elution concentration in batch 1 was higher than that in batch 2, while less volume was obtained in batch 1 than in batch 2; (3) 25 µL of each input lysate or flow through sample and 15 µL eluates were loaded in batch 1 while 10 µL of each Input lysate or flow through sample and 30 µL eluates were loaded in batch 2; and (4) blocked membrane (5% milk/tris-buffered saline with Tween-20, TBST) was incubated with the primary antibody against pan-14-3-3 (Santa Cruz Biotechnology, Inc; Cat. No. sc-133233) in batch 1, while blocked membranes were incubated with the primary antibody against pan-14-3-3 (ThermoFisher Scientific, Waltham, MA, USA; Cat. No. 51-0700) and PYGM (Proteintech, Rosemont, IL, USA; Cat. No. 19716-1-AP) in batch 2 (Appendix A). In spite of these differences between the two batches, the loading volume of IP and IgG from the same sample pool was identical, enabling comparison of protein abundance in IP vs. IgG as a validation of a specific protein binding to 14-3-3 in mouse heart.

### 3.8. 14-3-3 Interacting Proteins in the IP Samples Compared to the IgG Control Samples

Three statistical methods were used to determine the 14-3-3 interacting proteins (Appendix A).

1)TTEST: It is the most often used method to calculate the difference between two groups of samples. We adopted the paired t-test to calculate the fold-change, log2(fold-change), and *p*-value using the log2NSAF data. If there were equal to or greater than one sample with NA value for a certain protein, we curated the statistical results for the protein manually. At last, we adjusted the *p*-value using the BH method to obtain the FDR.2)QSPEC: [43] This method was based upon the hierarchical Bayes estimation of Generalized Linear Mixed effects Model (GLMM). This modeling strategy is proposed to be more powerful than calculating the signal-to-noise ratio type of differential expression test statistics. We adopted the pair alternating algorithm to calculate the fold-change, log2(fold-change), and FDR using the SpC data.3)Countdata: [41,42] This is a new R package to use the statistical method described in two published articles for mass spectrometry data. It is built based upon the beta-binomial model to analyze the spectral count data in label-free tandem mass spectrometry-based proteomics [42]. We adopted the paired comparison [41] to calculate the fold-change, log2(fold-change), and FDR using the SpC data. The normalized countdata was also used in the data distribution, sample distance and PCA analyses (Appendix AD–F).

The overlaps of results from the above three methods were analyzed in venn diagrams and the overlap between the results from countdata and QSPEC and within the cutoff (−log10(FDR) > 1 and log2(IP/IgG) > 0) were chosen for the subsequent analysis (Figure 2 and Appendix A). There are 52 proteins in this group. The log2(fold-change) values obtained from the QSPEC method were used in circos plot and correlation scatter plot (Figure 3).

### 3.9. Isolation of Adult Mouse Left Ventricular Myocytes and Immunofluorescence Staining

Following the previously described method [44], LV cardiomyocytes were isolated from 3-month old male C57 mice and cultured on laminin coated MatTek dishes in culture medium (composed of 5% fetal bovine serum, 47.5% 94 MEM, 47.5% modified Tyrode’s solution, 10 mM pyruvic acid, 4.0 mM HEPES, and 6.1 mM 95 glucose) for 1 h in a 5% CO_2_ atmosphere. HL-1 cells were cultured in Claycomb culture media and maintained in a 5% CO_2_ atmosphere.

For mitochondrial labeling in LV cardiomyocytes, cells were incubated for 30 min in CO_2_ incubator with MitoTracker Deep Red 633 (ThermoFisher: M22426, Molecular Probes) in the culture medium for following immunofluorescence staining, after which cells were fixed with ice-cold 100% methanol for 15 min at −20 °C. For mitochondrial labeling in HL-1 cell line, cells were fixed with 4% paraformaldehyde for 10 min, permeabilized with 0.1% Triton™ X-100 for 10 min. All cells were washed two times with PBS and blocked with 1% BSA for 1 h.

Independently from MitoTracker labeling, we used anti-TOMM20 and ribosome markers. Cells were fixed as follows: LV cardiomyocytes were fixed with ice-cold 100% methanol for 15 min at −20 °C. HL-1 cells (passage 56) were fixed for 20 min with 1:1 Acetone/Methanol fixative at −20 °C. All plates were washed twice with PBS and then incubated with 10% goat serum for 1 h to minimize nonspecific staining.

For following immunofluorescence staining, all samples were incubated at 4 °C overnight with primary antibodies against specific: (1) pan 14-3-3 antibody (H-8) (Santa Cruz: sc-1657; 1:50); Mitochondrial Marker (2) anti-TOMM20 antibody (Abcam: ab78547; 1:100); Ribosome Marker (3) anti-S6 Ribosomal Protein (5G10) antibody (CST: #2217; 1:100), (4) anti-RPS3A antibody (GeneTex: GTX105029; 1:100), and (5) RPS3 (LS-C497796; 1:100). Then, cells were washed three times with PBS and incubated with fluorescence-conjugated secondary antibodies (1:1000), Goat anti-Rabbit IgG (H + L) Cross-Adsorbed Secondary Antibody Alexa Fluor^®^ 568 conjugate (ThersmoFisher, # A-11011), for 45 min at 37 °C. Cell nuclei were labeled with DAPI (Sigma, D9542-1MG) and covered by SlowFade^®^ Gold Antifade Mountant (ThermoFisher, # S36938).

Cells were visualized by LSM 710 laser-scanning confocal microscope (Carl Zeiss, Jena, Germany) and images were captured using the Carl Zeiss Zen software (Zen 3.0 (black edition) Carl Zeiss Microscopy Gmbh2019 Germany).

### 3.10. Comparison of Proteins Identified in Our Interactome and Proteins in Annotation and Integrated Analysis (ANIA) Database

ANIA [45,46] (https://ania-1433.lifesci.dundee.ac.uk/prediction/webserver/index.py/getting_started, accessed on 5 October 2021, and https://github.com/JiahuaQu/Structural-Landscape-and-Deduced-Functions-of-The-Cardiac-14-3-3-Protein-Interactome/tree/main/Database/ANIA, accessed on 22 September 2022) is a public database, storing the analysis data of the 14-3-3 interactome, which integrates multiple data sets on 14-3-3-binding phosphoproteins. There are two kinds of information: (1) gold standard (GD) 14-3-3 binders have experimentally determined 14-3-3-binding sites, and (2) high throughput (HT) binders recorded in other public databases. Among the 52 interacting proteins, only 13 were mapped to the recorded proteins in ANIA (HT). The remaining 39 proteins may be non-phosphorylated proteins not recorded in ANIA or undiscovered interacting proteins, particularly in the mouse heart, due to technical or material limitations in previous studies. Therefore, the present study provides abundant proteins for further analyses beyond those showed there (Appendix A).

### 3.11. Comparison of Proteins Identified in Our Interactome and Proteins in BioPlex Database

The most recent BioPlex 3.0 [47] (https://bioplex.hms.harvard.edu/) includes nearly 120,000 interactions among nearly 15,000 proteins, and is one the most comprehensive experimentally derived model of the human interactomes. It records two types of protein–protein interactions: (1) in HEK293T cell-line (BioPlex 293T), and (2) in HCT116 cell-line (BioPlex HCT116). Because we identified 14-3-3β (*Ywhab*), 14-3-3ε (*Ywhae*), 14-3-3η (*Ywhah*), and 14-3-3θ (*Ywhaq*) in our cardiac 14-3-3 interactome, we searched the proteins binding to those four proteins in the BioPlex databases. The proteins identified in BioPlex and our interactome were compared in venn plot (Figure 2F), and the lists of proteins were in Appendix A.

### 3.12. Prediction of 14-3-3 Binding Sites via 14-3-3-Pred

14-3-3-Pred (http://www.compbio.dundee.ac.uk/1433pred/, accessed on 21 September 2021) [48] is a webserver that predicts 14-3-3-binding sites by combining predictions from three different classifiers: ANN, PSSM and SVM. The 39 proteins identified in the cardiac 14-3-3 interactome, but not in ANIA database, were queried using their UniProt IDs in 14-3-3-Pred. Information on the phosphorylation state of the respective Ser/Thr, in addition to prediction scores, is provided for each queried protein in the output table. We collated those tables and added columns to indicate the corresponding UniProt IDs, protein names and gene names (Appendix A).13. Protein–protein interactions (PPIs) analysis in STRING database.

STRING database [49] (version: 11.0; http://string-db.org/, accessed on 5 October 2021) was used to analyze the PPIs in order to detect all interactions between the interacting proteins. To portray the global landscape of the whole 14-3-3 interactome, all the 52 selected proteins were uploaded to the multiple proteins search tool in STRING database. The parameters were set as: (1) Network edges: confidence; (2) Line thickness: strength of data support; (3) Active interaction sources: all; (4) Minimum required interaction score: the default, medium confidence (0.400); (5) Kmeans Cluster Algorithm (*n* = 3) clustering. After calculation, a large and complex network with 51 14-3-3 interacting protein nodes and 65 edges was generated by STRING (Figure 4). In the network, the average node degree is 2.55, the local clustering coefficient is 0.452, and the PPI enrichment *p*-value is 6.04 × 10^7^. Three inter-connected subgroups were clustered and proteins in each subgroup were exported and listed in Appendix A. The features of the three clusters were analyzed using the 12 public databases: COMPARTMENTS from Subcellular localization (COMPARTMENTS) database, Component from Cellular Component (Gene Ontology) database, Function from Molecular Function (Gene Ontology) database, InterPro from Protein Domain and Features (InterPro) database, KEGG from Kyoto Encyclopedia of Genes and Genomes pathway (KEGG) database, Key word from Annotated Keywaords (Uniprot) database, NetworkNeighborAL from local network cluster (STRING) database, Pfam from Protein Domains (Pfam) database, Process from Biological Process (Gene Ontology) database, RCTM from Reactome Pathways database, SMART from Protein Domains (SMART) database, and WikiPathways from WikiPathways database (Figure 4, Appendix A and Appendix A).

### 3.13. Functional Analyses in clusterProfiler

The IDs of the 52 precleared 14-3-3 bound proteins were transformed into ENTREZ IDs in IPA. Their ENTREZ IDs were submitted to clusterProfiler (version: 4.0.5) [50] and analyzed using the method of overrepresentation enrichment analysis (ORA). The databases of biological process (GO_BP), cellular component (GO_CC) and molecular function (GO_MF) were selected for gene ontology (GO) analysis. The Kyoto Encyclopedia of Genes and Genomes (KEGG) database was selected for pathway analysis. The cutoff parameters were set as: pvalueCutoff = 0.05, pAdjustMethod = “BH”, and qvalueCutoff = 0.1. The obtained results were listed in Appendix A and displayed in bubble charts (Figure 5).

### 3.14. Visualization of GO Analysis Results after Redundancy Reduction in REVIGO

To visualize the GO analysis results, all the GO terms obtained from clusterProfiler were imported into REVIGO [51] (http://revigo.irb.hr/, accessed on 7 October 2021). To reduce redundancy, the default parameters in REVIGO were applied to generate bubble charts (Appendix A and Appendix A). Bubble charts after redundancy reduction depict clusters in a two dimensional space derived by applying multidimensional scaling to a matrix of the GO terms’ semantic similarities [52,53]. Consequently, the bubbles’ closeness on the plot should closely reflect their closeness in the GO graph structure, i.e., the semantic similarity.

### 3.15. Canonical Pathway Analysis and Visualization in Ingenuity Pathway Analysis (IPA)

The 52 processed proteins were uploaded into the latest released version of IPA (QIAGEN, October 2021) [54]. The Ingenuity knowledge base (gene only) was selected as the reference set. All Ingenuity supported third party information and Ingenuity expert information were adopted as the data sources, and only experimentally observed relationships were selected to increase confidence. After screening, all the 52 proteins could be recognized by IPA and were also matched to Mouse ENTREZ IDs. All the enriched pathways and from which metabolic pathways and cardiovascular signaling pathways are listed in Appendix A. The top 30 pathways in which our cardiac 14-3-3 proteins were enriched were displayed (Figure 6 and Appendix A).

### 3.16. Protein Localization Analysis in Integrated Mitochondrial Protein Index (IMPI), MitoCarta and AmiGO Databases

Proteins identified in our cardiac 14-3-3 interactome were compared with the corresponding proteins within the following three mitochondrial protein databases to predict the subcellular localization (Figure 7A).

IMPI (version: Q2 2018; http://www.mrc-mbu.cam.ac.uk/impi, accessed on 5 October 2021) is embedded in MitoMiner [55]. and collects genes that encode proteins with strong evidence for cellular localization within the mammalian mitochondrion. MitoCarta [56,57] (version: 2.0; https://www.broadinstitute.org/scientific-community/science/programs/metabolic-disease-program/publications/mitocarta/mitocarta-in-0, accessed on 5 October 2021) is an inventory of genes that encode mitochondrial proteins. The GO term of mitochondrion in AmiGO [58,59,60] (version: 2.5.12; Last file loaded on 18 April 2019; http://amigo.geneontology.org/amigo/search/bioentity?q=mitochondrion, accessed on 5 October 2021) represents the mitochondrial localization in some extent.

### 3.17. Integration of Information about Proteins Identified in Our Interactome and Summarized in the Review about Cellular Energy Metabolism

We compared our cardiac 14-3-3 interactome to Figure 1 in a review of the involvement of 14-3-3 proteins in core metabolic pathways and regulatory signaling pathways by Kleppe et al. [7]. The detailed information about the involved pathways was in Appendix A and the results were visualized in Figure 8. The components identified in our cardiac 14-3-3 interactome results were mapped to Figure 1 in Kleppe R. et al. [7].

### 3.18. Data Process and Graphics Production

Transcriptomic data were processed in Partek^®^ Flow^®^ software, version 9.0 Copyright©; 2020 Partek Inc., St. Louis, MO, USA, by utilizing the computational resources of the NIH HPC Biowulf cluster (http://hpc.nih.gov, accessed on 15 October 2021). All the raw data were precleared and processed using RStudio (version: 1.4.1717) in R language (version: 4.1.1), in which the tidyverse and ggvenn packages were applied. In addition, Microsoft Excel (version: 2021) and Adobe Illustrator (version: CC 2021) were also used for statistics and graphics. Circos plot was generated using the public Circos software (Vancouver, Canada) in Perl language [61].

## 4. Results

### 4.1. 14-3-3 Isoform Protein Expression in Mouse Heart

To begin, we interrogated a public transcriptomic dataset (via RNA-seq) of three-month old wildtype C57BL/6J male mouse hearts [39] in order to identify the RNA expression pattern of 14-3-3 isoforms: *Ywhab*, *Ywhae*, *Ywhag*, *Ywhah*, *Ywhaq* and *Ywhaz* (Figure 1A). The abundance pattern of the six 14-3-3 protein isoforms (14-3-3β, 14-3-3ε, 14-3-3γ, 14-3-3η, 14-3-3θ, and 14-3-3ζ) identified in another public proteomic dataset of three-month old wildtype C57BL/6J male mouse hearts matched the 14-3-3 RNA expression pattern: cardiac 14-3-3ε proteins displayed the highest abundance, followed by 14-3-3γ, while 14-3-3θ displayed the lowest abundance (Figure 1B). Although we confirmed the existence of the 14-3-3 proteins in the mouse heart, we still do not know their interaction. Therefore, we continued to conduct the mass spectrometry to obtain the global interaction of 14-3-3 proteins and their binding partners in the three-month old C57BL/6J male mouse hearts (Figure 1C,D).

### 4.2. A Global Landscape of 14-3-3 Interaction Network in Mouse Heart

As 14-3-3 proteins usually function as heterodimers [62,63,64], we utilized the pan-14-3-3 protein antibody and its isotype IgG control antibody in co-IP of mouse LV lysates. We identified 912 proteins in the paired design experiment where we pooled three mouse LVs into a pool and incubated the same amount of lysate from the same pool with the pan-14-3-3 protein antibody and its isoform IgG control antibody, respectively. After running the gel to observe the pulled down proteins in the lysate (Appendix AA), we excised each lane of the SDS-PAGE gel into ten equally sized segments for the mass spectrometry conduction, which produced the protein spectral counts (SpC). We normalized the count in three forms: Normalized Spectral Abundance Factor (NSAF) and its log2 transformed value, log2NSAF (Appendix AB,C), and normalized count in R package, countdata (Appendix AD). The normalized count in countdata was also used in the sample distance and PCA analyses. Both of the sample distance analysis (Appendix AE) and principal component analysis (PCA) (Appendix AF) showed that the IP against pan-14-3-3 (IP samples) and against the isotype IgG (IgG samples) were distinct from each other. The intense signal for 14-3-3 indicated a robust capture of 14-3-3 proteins (Appendix AA,B), so the data were valid for the following analysis.

The log2NSAF data were used in the TTEST statistics where 30 proteins were identified as 14-3-3 interacting proteins in the IP samples compared to the IgG control samples (Figure 2A). Because the label-free mass spectrometry data were regarded closer to other distributions, two newer methods, QSPEC and countdata, were adopted, which were built upon the hierarchical Bayes estimation of Generalized Linear Mixed effects Model (GLMM) and the beta-binomial model respectively. There were 173 (Figure 2B) and 59 (Figure 2C) proteins identified respectively. To select the most reliable binding proteins to 14-3-3, we took the intersection of the results from the different methods. There were only 19 overlapped proteins across the results from the three methods while 52 proteins from countdata and QSPEC, which were considered more suitable for the label-free mass spectrometry data [41,43]**.** Four 14-3-3 isoforms, 14-3-3β (*Ywhab*), 14-3-3ε (*Ywhae*), 14-3-3η (*Ywhah*), and 14-3-3θ (*Ywhaq*), were in this group of 52 proteins. To validate the interactors, we also performed Western blotting following co-IP. In addition to 14-3-3, we applied the co-IP experiment to a single candidate, PYGM, which catalyzes the rate-limiting step in glycogen catabolism and plays a central role in maintaining cellular and organismal glucose homeostasis. The Western blots showed the visible enrichment of PYGM in 14-3-3 Ab IP elution, but not in the Isotype Ab IP elution at all (Appendix AC,D). These results supported the efficacy of the pan-14-3-3 antibody and the co-IP experiment (Appendix A).

By combining the present 14-3-3 interactomic dataset with the above mentioned two public omics datasets (Figure 1A,B), we derived the circos plot integrating multiple omics datasets (Figure 3A). In the center of the circos plot, the links represent the possible bindings from one of the four 14-3-3 isoforms, 14-3-3β (*Ywhab*), 14-3-3ε (*Ywhae*), 14-3-3η (*Ywhah*), and 14-3-3θ (*Ywhaq*), to any other protein. The location of a protein is corresponding to the location of the encoding gene in the ideogram in the outermost layer (Figure 3A). From the different circles and the correlation scatter plots, we observed that the level of 14-3-3 interactions is independent of that of cardiac RNA or protein (Figure 3B–D). Particularly, three genes, Impdh2, Vwa8 and Sugt1, showed high level of expression in 14-3-3 interactome while their mRNA and protein expressions were relatively low (Figure 3A).

The ANnotation and Integrated Analysis (ANIA) database records many interactions, discovered mainly in human cell lines, of phospho-proteins and 14-3-3 proteins [45,46]. There are only 13 proteins that were identified in our cardiac 14-3-3 interactome and also recorded in the ANIA HT database (Figure 2E and Appendix A). The cardiac 14-3-3 PPIs provided 39 interacting proteins beyond those recorded in the ANIA database (blue in Figure 2E). We also searched the proteins interacting with 14-3-3 that were recorded in BioPlex database [47], and also only got a small number of overlapped proteins identified in the two human cell-lines, HEK293T and HCT116, and our mouse cardiac 14-3-3 interactome (Figure 2F and Appendix A). The thirty-nine 14-3-3 interacting proteins within our samples that did not overlap with those in ANIA database (blue in Figure 2E) may be in non-phosphorylated states, or may be specific to mouse LV, because we did not observe any phosphorylation state of binding sites on these 39 proteins that were predicted by 14-3-3-Pred [48] (A webserver to predict 14-3-3-binding sites in proteins) (Appendix A). Therefore, it is of great scientific interest to study the roles of the 14-3-3 interactome in the mouse heart.

We combined multiple bioinformatic tools and approaches to comprehensively dissect the structure and deduce functions of cardiac 14-3-3 protein–protein interactions (PPIs) (Figure 1E,F). For structure, we used STRING to analyze the entire 14-3-3 PPIs.

### 4.3. STRING PPIs Analysis

The analysis in STRING unveiled the global landscape of the cardiac 14-3-3 interactome (Figure 4). There were 51 nodes recognized in the highly robust network, with an average node degree of 2.55; and 65 edges, representing the physical PPI, were in the network, with a PPI enrichment *p*-value 6.04 × 10^−7^. Using the kmeans (*n* = 3) algorithm, we deduced the internal relations of the structural aspects of the 14-3-3 interactome, which was consisted of three main clusters.

### 4.4. STRING Cluster 1: The Cardiac 14-3-3 Mitochondrial and Metabolic PPI

In cluster 1 (green color in Figure 4A and Appendix A), there were 21 nodes and 27 edges. The average node degree was 2.57 and the average local clustering coefficient was 0.405. The PPI enrichment *p*-value was less than 1.0 × 10^−16^, indicating a significantly interacting network (Figure 4B). To study the roles of the large PPI network, we employed 12 databases to enrich the functions of cluster 1 into 12 categories showed in 12 different colors (Appendix A). The 12 categories covered biological process, cellular component, molecular function, signaling pathway, protein domain, etc. With the cutoff FDR < 0.05, if there were equal to or greater than five terms enriched in one category, only the top five terms, ranked by −log10FDR, were shown; otherwise, all terms were shown. Intriguingly, mitochondrion and mitochondrial function were mostly enriched across the 12 databases.

The term, mitochondrion, was enriched significantly in Component, Keyword and COMPARTMENTS. Other mitochondrial components including mitochondrial membrane and mitochondrial envelope were also enriched and ranked on the top of the bar plot. Particularly, some subunits in mitochondrial complex I, Ndufb6, Ndufb9 and Ndufb10, were in cluster 1. In addition, Vwa8 (Von Willebrand A Domain-containing Protein 8), a AAA + ATPase that is associated with the matrix face of the inner mitochondrial membrane [65], was not only in cluster 1 but also a representative molecule in the above Circos plot (Figure 3A).

The mitochondrial function, citric acid (TCA) cycle, appeared in the categories of KEGG, WikiPathways and RCTM, as well as oxidative phosphorylation in WikiPathways. Because mitochondrion is also the metabolic center, many metabolism related terms were enriched consistently from cluster 1, suggesting that the oxidation-reduction process happened in mitochondria during metabolism results in many metabolites and energy. Therefore, acetylation, an organic esterification reaction with the metabolite, acetic acid, was enriched in Keyword.

### 4.5. STRING Clusters 2: The Cardiac 14-3-3 Homeostatic PPI

There were 13 nodes and 11 edges in cluster 2 (red color in Figure 4A and Appendix A). The average node degree was 1.69 and the average local clustering coefficient was 0.436. The PPI enrichment *p*-value was 2.38 × 10^−5^, indicating a smaller interacting network than cluster 1 (Figure 4B). There were two major types of proteins in this cluster, proliferation and degradation proteins. For example, Impdh2 (Inosine-5’-monophosphate dehydrogenase 2) mainly works as the rate-limiting enzyme in the de novo guanine nucleotide biosynthesis to catalyze the conversion of inosine 5’-phosphate to xanthosine 5’-phosphate, and is thus important in the regulation of cell growth [66,67,68]. The enriched key words of protein biosynthesis and elongation factor could also contribute the proliferation and growth.

Another representative protein was Sugt1(Protein SGT1 homolog) that is predicted to interact with many chaprones like heat shock protein 90 in STRING (https://version11.string-db.org/cgi/show_network_section.pl?taskId=5WOyo17Q1d4t, accessed on 5 October 2021) and is supposed to play a role in ubiquitination and subsequent proteasomal degradation of target proteins in UniProtKB/Swiss-Prot (https://www.uniprot.org/uniprot/Q9CX34, accessed on 5 October 2021). Both of the two proteins, Impdh2 and Sugt1, were representative molecules in the Circos plot, and their binding to 14-3-3 was independent of their RNA or protein expression in the mouse heart (Figure 3). Therefore, the proteins in this cluster contribute to the maintenance of cardiac homeostasis.

The cluster feature analysis highlighted the Ku10 and Ku80 related terms in the bar plot (Appendix AB). Ku is a dimeric protein complex, which consists of two polypeptides, Ku70 (XRCC6) and Ku80 (XRCC5) in eukaryote. The complex is required for DNA repair via non-homologous end joining (NHEJ) pathway [69]. Consistently, NHEJ was also enriched here.

Taken together, 14-3-3 interacting proteins in cluster 2 could consume the metabolites and energy produced in cluster 1 for protein biosynthesis while degrade impaired proteins and repair other DNA damages, consequently maintaining the cardiac homeostasis.

### 4.6. STRING Clusters 3: The Cardiac 14-3-3 Cytoskeleton PPI

The remaining 17 proteins were connected by 25 edges in cluster 3 (blue color in Figure 4A and Appendix A). The PPI has the average node degree 2.94, the average local clustering coefficient 0.618, and enrichment *p*-value 6.52 × 10^−8^ (Figure 4B). The enriched processes and functions are mainly in the cytoplasm and cytoskeleton.

All the four 14-3-3 isoforms were clustered in this group, so many 14-3-3 related terms were enriched from the proteins in cluster 3 (Appendix AC). 14-3-3 proteins were known to bind various partners inside a cell, and thus a lot of binding related terms were also appeared in this cluster, as well as some junction related terms. The regulated ion channel binding and PI3K-Akt signaling pathway might result in calcium regulation in cardiac cells.

In line with the features in cluster 1 and 2, cluster 3 also contained the key word, acetylation as well as ubiquitin-like (Ubl) protein conjugation [70] and thus could contribute to the cardiac homeostasis maintenance in addition to cluster 2.

### 4.7. Functional Analyses of 14-3-3 Interacting Proteins in Mouse Heart

The cluster analyses consisting of the 51 proteins in the three clusters (Figure 4) occupy 98% of the entire 14-3-3 interactome. We next conducted stringent functional analyses of the entire interactome, taking into consideration of the interacting proteins in three clusters as a whole, where we applied multiple additional bioinformatic methods and algorithms: Gene Ontology (GO) analysis, Kyoto Encyclopedia of Genes and Genomes (KEGG) pathway analysis and Ingenuity Pathway Analysis (IPA) (Figure 1F). The GO analysis was classified into three categories, Biological Process (BP), Cellular Component (CC) and Molecular Function (MF).

First, all the 52 proteins were imported into Rstudio for functional enrichment using R package, clusterProfiler. The results were displayed in bubble charts (Figure 5 and Appendix A). Then, the lists of BP, CC and MF were uploaded into REVIGO online tool to reduce the redundancy and visualize the correlation of terms in 2-D space (Appendix A and Appendix A). At last, to further interpret the complex cardiac 14-3-3 interaction network, we made full use of the manually curated content of the Ingenuity Knowledge Base, a unique advanced feature within Ingenuity Pathway Analysis (IPA) software that accounts for the identification of the most significant pathway enrichments (Figure 6 and Appendix A).

In BP analysis, metabolism and energy related processes were ranked on the top (Figure 5A). After the redundancy reduction, some representative processes were localized in the hub of some clusters in the 2-D space (Appendix AA), including generation of precursor metabolites and energy, carbohydrate catabolic process, NAD metabolic process, NADH metabolic process and ATP metabolic process. The regulation of metabolism was supported by the results in MF analysis, such as ATPase activity, ATP hydrolysis activity, AMP binding, GTPase activity, GTP binding, carbohydrate kinase activity, carbohydrate binding, and oxidoreductase activity (Figure 5C and Supplementary S4C), and the results in KEGG analysis, such as carbon metabolism and TCA cycle (Figure 5D). There were a large number of canonical pathways computed in IPA (Figure 6A), among which 12 metabolic pathways were enriched significantly with −log10(*p*-value) > 1.3 (Figure 6B). Because many metabolic processes happen in mitochondria, the process of mitochondrion organization also occurred in BP analysis (Appendix AA). Consistently, mitochondria, oxidoreductase complex and respirasome were enriched in CC analysis (Figure 5B and Appendix AB). The enriched canonical pathway, mitochondrial dysfunction, indicated that the abnormality of the 14-3-3 interactome might result in mitochondrial dysfunction in the mouse heart (Figure 6A).

In accord with the heart tissue from which the samples were, regulation of the force of heart contraction was enriched in BP analysis (Appendix AA), and many cardiac cellular components were enriched in CC analysis, such as Z disc, I band, sarcomere, myofibril, contractile fiber and intercalated disc (Figure 5B and Supplementary S4B). Therefore, nine cardiovascular signaling pathways were in the IPA result (Figure 6C), indicating the importance of the 14-3-3 PPIs, especially the interactions to Col1a2, F13a1, Ppp1r3a, Prkag1, Ryr2, Arhgef5, Cdc42 and Rac1, in the regulation of cardiac functions (Appendix A).

Unexpectedly, five 14-3-3 interacting proteins, Atp1a2, Cdc42, Dlst, Tcp1 and Tufm, were localized in myelin sheath shown in CC analysis (Appendix AB). The myelin sheath is a fatty insulating sleeve that surrounds and protects nerve cells. Its recognized main function is to buffer the disturbances of other electrical impulses in order to expedite the transmission of nerve signals [71,72]. The functional roles of these proteins may be also involved in multiple metabolic processes and catalytic activity (Appendix A).

Taken together, the GO, KEGG and IPA functional analyses (Figure 5 and Appendix A), consistent with STRING analysis (Figure 4 and Appendix A), suggested the involvement of 14-3-3 interactome in the regulation of mitochondrial function, metabolism, molecular binding and cellular junction, and cardiac function and homeostasis.

### 4.8. Localization Validation of 14-3-3 Interacting Proteins in the Mouse Heart

We had already observed the enrichment of mitochondria related terms in the above bioinformatic analysis. Next step, we combined three credible mitochondrial databases (Integrated Mitochondrial Protein Index (IMPI), MitoCarta and AmiGO). Of the 52 cardiac proteins that bind 14-3-3s, half were mapped to mitochondrial proteins recorded in the three databases (Figure 7A), supporting the localization mainly in mitochondria. To validate the mitochondrial localization, we conducted immunofluorescence staining assays to detect the subcellular localization of 14-3-3 proteins (Figure 7B). To increase generality to more than one type of cardiomyocyte, we used two types of cardiomyocytes, HL-1 cell line and primary mouse LV cardiomyocytes; and to increase credibility, we adopted two different mitochondrial indicators, TOMM20 and MitoTracker. Results demonstrated the definite colocalization of 14-3-3 proteins in mitochondria.

Because the protein biosynthesis and elongation were also enriched in the above bioinformatic analysis (Figure 5C and Appendix A) and protein synthesis is processed in ribosome, we speculated that 14-3-3 could also interact with proteins in ribosome. To test it, we used three antibodies to target ribosomal proteins (RPS3A, RPS3 and RPS6) in immunofluorescence staining assays in primary mouse LV cardiomyocytes and HL-1 cell line (Figure 7C).

The strong signal in cardiomyocytes was in line with results in CC analysis, i.e., Z disc, I band, sarcomere, myofibril, contractile fiber and intercalated disc (Figure 5B and Appendix AB). Compared with the immunofluorescence staining images for mitochondria (Figure 7B), ribosome labeling was coincided with the I-bands and on both sides of z disc, whereas mitochondria in young mice were oriented along the long axis, in line with previous reports [73,74], Although we also observed the colocalization of 14-3-3 proteins with ribosomes, more 14-3-3 proteins were distributed along the long axis in myocytes, indicating stronger colocalization with mitochondria than ribosomes.

### 4.9. Mapping the Cardiac 14-3-3 Interactome to Cellular Metabolic Network

Because mitochondrial is the metabolic hub, protein synthesis and DNA/protein repair consume large amount of energy, and ubiquitylation and protein degradation recycle metabolites, we proposed the hypothesis that the cardiac 14-3-3 interactome plays a role in the cellular metabolism. We learned from a well-organized review about the 14-3-3 interactions to proteins widely involved in core metabolic pathways and regulatory signaling pathways [7]. However, all the evidence in that review was collected from studies in non-cardiac cells or tissues due to the lack of research in the cardiac 14-3-3 interactome [7]. We annotated metabolism related BP, CC, MF, KEGG and IPA terms identified in the cardiac interactome by adding stars to Figure 1 in the review by Kleppe R. et al. [7] which had previously linked the 14-3-3 PPIs to the metabolism (Figure 8). The mapped proteins were classified into two categories: (1) yellow stars represent proteins involved in energy production, e.g., glycolysis, Krebs (TCA) cycle, and β-oxidation; (2) green stars represent proteins involved in energy consumption, including translational machineries, elongation factors and other regulators, e.g., insulin signaling, apoptosis, autophagy, amino acid biosynthesis, TOR signaling, AMPK signaling, and MAPK/ERK signaling.

In the category of energy production, process of glycolysis provides pyruvate and Acetyl-CoA as the resource for Krebs cycle (TCA cycle) and β-oxidation as well as acetylation. As to the category of energy consumption, 14-3-3 interacting proteins are related to mRNA translation. Other regulators, such as MAPK/ERK kinase, PKA and AMPK, play important roles in the regulation of TOR signaling pathway to regulate translation. The processes of apoptosis and autophagy promote the metabolite turnover. Annotations not only prompted the idea that cardiac 14-3-3 PPIs regulate the same mitochondrial functions and cellular energy metabolism as in non-cardiac tissues, but also added knowledge to potential 14-3-3 regulation mechanisms in heart, especially in energy homeostasis.

## 5. Discussion

14-3-3 proteins bind to a large number of proteins in non-cardiac cells and tissues to modulate the functions of metabolic enzymes and impact metabolic processes and other signaling pathways [7]**.** However, neither the cardiac 14-3-3 protein interactome nor the numerous and complex potential roles of 14-3-3 PPIs in energy supply/demand or protein quality control in heart had been catalogued. To this end, we evaluated the expression of 14-3-3 RNA and protein isoforms in mouse heart and conducted co-IP and MS to identify mouse LV proteins that bind to 14-3-3 proteins, i.e., the cardiac 14-3-3 protein interactome. We designed the paired experiment to identify the significantly binding proteins in the IP sample with the pan-14-3-3 antibody compared to the IgG control sample with the isotype IgG antibody. To improve the specificity, we adopted three statistical methods built upon three models and took the intersection to obtain 52 14-3-3 binding partners in the mouse heart. Multiple bioinformatic tools and methods were employed to clarify the structure of the complex 14-3-3 interactome and then to deduce the cellular components, biological processes, molecular functions and pathways in which the cardiac 14-3-3 PPIs are involved. Our results show that by binding to various targets spread throughout the cell, 14-3-3 proteins are involved in multiple signaling pathways, especially those related to metabolism, protein quality control and cardiac homeostasis. Therefore, the global network of 14-3-3 PPIs in mouse LV senses and responds to multiple stimuli and stresses to coordinate signaling pathways that are involved in the maintenance of cardiac energy production and distribution to cell energy consumers, e.g., cardiac muscle contraction and cellular homeostasis (Figure 9).

### 5.1. The Cardiac 14-3-3 PPIs and Energy Production

Mitochondria integrate the metabolism of three major nutrients, glucose, lipids and proteins, to produce energy. Our results demonstrate that 14-3-3 proteins localize within cardiac mitochondria. A substantial number of 14-3-3 bound proteins (26 proteins) were mapped to the mitochondrial protein databases (Figure 7A). The enrichment of mitochondrion for cardiac 14-3-3 interacting proteins occupied the top of the GO CC list (Figure 5B and Appendix AB), and 14-3-3 interactome was enriched in various aspects of cardiac mitochondrial metabolism, including glucose, fatty acid and amino acid metabolism (Figure 5, Figure 6 and Appendix A), in agreement with other reports in non-cardiac cells and tissues [7]**.** Specifically, many proteins identified in the cardiac 14-3-3 interactome have been mapped to core metabolic pathways and regulatory signaling pathways in non-cardiac cells or tissues (Figure 8) [7]**.** The integrated map in Figure 8 annotated with enriched terms of the present study illustrates that the cardiac 14-3-3 PPIs modulate the same 14-3-3 target proteins discovered previously in non-cardiac cell/tissue, related to many energy production and consumption processes including glycolysis, β-oxidation, Krebs cycle, and mRNA translation. Importantly, glucose and fatty acid metabolism was mostly affected, as TCA cycle, carbon metabolism, acetylation, acyl-CoA hydrolysis and glutaryl-CoA degradation were significantly enriched in bioinformatic analysis (Figure 5, Figure 6and Appendix A). Because heart uses fatty acid as the main energy source, regulation of fatty acid β-oxidation by 14-3-3 interactome likely plays an important role in triaging energy production to maintain heart health. For example, Hsd17b10 (3-hydroxyacyl-CoA dehydrogenase type-2), involved in the pathways of fatty acid, branched-chain amino acid and steroid metabolism [75,76,77], is essential for mitochondrial structural and functional integrity [78]. It was identified as one 14-3-3 interacting protein in the mouse heart and clustered in cluster 1. An interaction between Hsd17b10 and 14-3-3ε was inferred in hepatocellular carcinoma cells, but not confirmed [79]. The function of its binding to 14-3-3 proteins in the heart remained to be studied in further.

Some cardiac 14-3-3 binding proteins are, in fact, proteins localized in mitochondrial complexes, especially those protein subunits in mitochondrial complexes I, Ndufb6, 9 and 10. The ATPase activity was also enriched to coordinates ATP metabolism. In fact, our results suggest that 14-3-3 binding may have a greater impact on those proteins in cardiac mitochondrial complex I rather than other complexes. For instance, the significant enrichments of mitochondrial complex I (also called NADH dehydrogenase complex), NADH and NAD metabolic processes, and mitochondrial organization indicate that 14-3-3 proteins play an important role in the organization of complex I. Because mitochondrial complex I is the first step of oxidative phosphorylation and is the largest complex in the respiratory chain, 14-3-3 interactions with complex I proteins imply that the 14-3-3 PPIs regulate cardiac mitochondrial metabolism. A recent study reported a survival strategy of oocytes by suppressing mitochondrial complex I [80], but the deep molecular mechanism remains to be clarified. As mitochondria are involved in multiple systems, the mechanism discovered in the present study may be extensively applied into other systems. In addition to subunits in mitochondrial complex, other factors regulating mitochondrial functions were also identified in the cardiac 14-3-3 interactome, e.g., Dap3 (28S ribosomal protein S29, mitochondrial) regulating mitochondrial protein synthesis, Mfn1 (Mitofusin-1) regulating mitochondrial fusion, Letm1 (Mitochondrial proton/calcium exchanger protein) regulating proton/calcium transportation, etc., indicating that 14-3-3 proteins affect the mitochondria widely. This idea is supported by the 14-3-3 PPIs’ involvement in enzyme and kinase activities and oxidation-reduction process (Figure 5C and Appendix AA). Moreover, the 14-3-3 interactome not only likely regulates oxidation phosphorylation, but also acetylation, thus providing metabolites for other biological process and protein modification (Figure 6B and Appendix A). Thus, the cardiac 14-3-3 PPIs regulate mitochondrial functions and metabolism to modify the energy production in cells for various biological processes and cellular health in heart.

### 5.2. The Cardiac 14-3-3 PPIs and Energy Consumption

Cardiomyocyte contraction consumes large amount of energy produced in mitochondria, so 14-3-3 binding partners were involved in cardiac muscle contraction process (Appendix AA). Further, protein synthesis is the most energy-consuming cellular process [81] and interaction of 14-3-3s with numerous proteins involved in processes that regulate protein synthesis suggests that 14-3-3 PPIs coordinate energy production and energy consumption by protein synthesis. Dap3, a subunit in ribosome, the structural protein of translational machine, was identified as cardiac 14-3-3 binding partners, as were proteins that regulate elongation, Tufm (Elongation factor Tu, mitochondrial) and Eef1d (Elongation factor 1-delta). Studies in non-cardiac cells indicate that 14-3-3σ regulates mitotic translation via binding to a variety of translation/initiation factors, including eukaryotic initiation factor 4B, in several tumor cell lines, U2OS (human bone osteosarcoma cell line), HeLa (human cervical cancer cell line) and HCT116 (human colorectal carcinoma cell line) [82]. The annotated 14-3-3 interaction network also raises the importance of 14-3-3 PPIs in regulating energy production and energy consumption (Figure 8). 14-3-3 proteins serve as the triage that senses the nutritional signals. When there are enough nutrients or in emergency, 14-3-3 can bind metabolic proteins to produce energy for proteins synthesis that improve cardiac contraction to respond to stress.

Protein quality control, consisting of protein synthesis and degradation, is important to the health [83]. Exposure to internal or external stress often impairs protein folding. When damaged, unfolded proteins aggregate within the cell to interrupt crucial cellular functions. Therefore, the balance between the protein synthesis and protein degradation is key to maintaining cardiac proteostasis. Proteins involved in protein degradation, e.g., autophagy and apoptosis, were enriched in the cardiac 14-3-3 interaction network (Appendix A). Consistently, eight proteins (Ywhab, Kif5b, Vim, Map4, Ywhae, Rac1, Mfn1 and Ywhaq) were involved in Ubl conjugation (Appendix AC). In addition, Sugt1 is predicted to play a role in ubiquitination and subsequent proteasomal degradation of target proteins. It was identified as one 14-3-3 interacting protein in the mouse heart and clustered in cluster 2. An interaction between Hsd17b10 and 14-3-3ζ was inferred in mouse testis, but not confirmed [84]. A role of 14-3-3 PPIs in recognition of unfolded proteins and regulation of the unfolded protein response is suggested by the enrichment of chaperone binding and heat shock protein binding (Appendix AC and Appendix A). The mitochondrial unfolded protein response of damaged mitochondrial proteins maintains normal mitochondrial homeostasis and perhaps extends the lifespan [85,86]. Thus, the intimate relationships between the cardiac 14-3-3 PPIs with mitochondrial proteins and proteins that regulate proteostasis suggest a rationale of targeting 14-3-3s to promote cardiac health.

Our results suggest important roles of 14-3-3 proteins and the 14-3-3 PPIs in maintaining the mouse heart metabolic and protein homeostasis and perhaps even the whole body homeostasis (perhaps via myelin sheath and various signaling pathways). Therefore, manipulation of the 14-3-3 bound mitochondrial proteins and cardiac 14-3-3 PPIs may be leveraged to manage cardiac metabolic diseases as they are now being manipulated in novel cancer therapeutics [22,23,24].

### 5.3. Limitations

As the proteins were extracted from LV lysates, there is the possibility that some of our identified 14-3-3 interacting proteins were non-specific. Therefore, we performed immunofluorescence staining assays with different antibodies in more than one cell type to prove the interaction of 14-3-3 proteins with mitochondrial proteins and ribosomal proteins. We also performed co-IP to pan-14-3-3 followed by Western blot to verify cardiac 14-3-3 interactors. Many additional molecular biological experiments, e.g., bidirectional co-IP and Western blot, fluorescence resonance energy transfer, or transmission electron microscope, are required to be implemented before confirming the specific interactions between 14-3-3 proteins and their binding clients. Importantly, expressing a fusion protein of a 14-3-3 protein and a proximity-based ligase, such as TurboID [87], in the future studies would enable the capture of *dynamic* PPIs, perhaps being more important than the static immunoprecipitation results performed here and in most studies.

### 5.4. Therapeutic Potential

14-3-3 protein binding can regulate expression of other molecules [88] and modify interaction networks. [8] Many 14-3-3 and 14-3-3 PPIs inhibitors [89,90,91] and stabilizers [92,93,94,95] have been regarded as potential therapeutic targets in other systems than heart [22,23]. Small molecules, peptides and natural products that target 14-3-3 or 14-3-3 PPIs have been applied as 14-3-3 and 14-3-3 PPI modulators [96,97,98]. For example, one benefit of inhibition of 14-3-3 is to overcome drug resistance in cancer [18]. In the future, such therapeutic approches can be applied to physiological or pathological conditions of the heart, in hope of discovering new and effective therapies to improve cardiac health.

Our results unveil an extremely wide and complex 14-3-3 interaction network within mouse heart that appears to be closely linked to mitochondrial cellular components and metabolism. This 14-3-3 interaction network links energy production to major energy consuming processes that regulate cardiac contraction and protein quality control. The cardiac 14-3-3 interactome is a likely candidate (similar to an air traffic controller) to acutely deal with acute intracellular metabolic or ionic imbalance that emerges within the diurnal cycle of complex heart cells or in response to myocardial ischemia. Thus, the cardiac 14-3-3 PPIs are likely candidates to coordinate the energy production and consumption to maintain cell energy supply/demand, required for optimal cardiac contraction and proteostasis (Figure 9). The present study revealed the structure and function of 14-3-3 interactome in untreated/natural condition, laying the foundation to unravel the therapeutic potentials in pathological conditions.

In summary, our comprehensive structural and functional analysis provides an exciting and novel paradigm for investigating PPIs in mouse heart. Our comprehensive analysis of cardiac 14-3-3 PPIs affords enormous opportunities for future research focusing on mechanisms how a specific 14-3-3 PPI in heart regulates specific pathways in response to intracellular or environmental stresses. In the context, dysfunctions within 14-3-3 PPIs or the abnormalities in 14-3-3 bound proteins could result in intracellular molecular disorders that lead to severe diseases that are predominantly metabolic in nature, and increase in incidence with advancing age, e.g., age- and metabolism-related cardiovascular diseases. Therefore, the regulation of the 14-3-3 interaction network is undoubtedly a therapeutic target in metabolic and proteostatic aspects of cardiovascular disease states, as it already is in cancer therapy [22,23,24].

## Figures and Tables

**Figure 1 cells-11-03496-f001:**
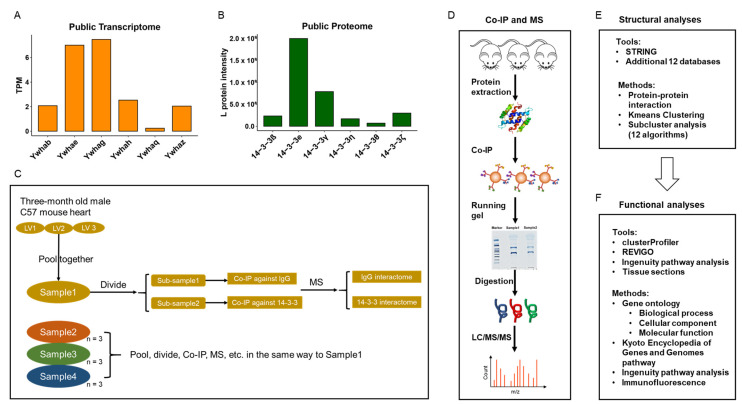
Conduction of 14-3-3 interactome in the mouse heart. (**A**) RNA expression (TPM) of 14-3-3 isoforms in three-month old wildtype C57 male mouse whole heart, processed from raw data of a public transcriptome (BioProject: PRJNA264807; Series: GSE62689). (**B**) Protein abundance of 14-3-3 isoforms in three-month old wildtype C57 male mouse heart, processed from L protein intensity of a public SILAC proteome. The gene/RNA name and corresponding protein name of each isoform are: (1) *Ywhab*, 14-3-3β; (2) *Ywhae*, 14-3-3ε; (3) *Ywhag*, 14-3-3γ; (4) *Ywhah*, 14-3-3η; (5) *Ywhaq*, 14-3-3θ; (6) *Ywhaz*, 14-3-3ζ. (**C**) The experiment design. Every three mouse left ventricles were randomly selected and pooled into a pool sample, which were then divided into two sub-samples for co-IP against 14-3-3 or isotype IgG in pair. (**D**) The workflow of co-immunoprecipitation (co-IP) and mass spectrometry (MS) for obtaining the 14-3-3 interacting proteins in mouse heart. E-F. Bioinformatic analyses tools and methods for (**E**) structure and (**F**) function of the mouse LV 14-3-3 PPI. As to structural analyses, the methods of protein–protein interaction are used to generate the overall interaction network and the kmeans method is used to cluster proteins in groups in STRING. The features of the clusters were analyzed with the knowledgebase in 12 public databases. As to functional analyses, the R package, clusterProfiler, was used to analyze GO categories and KEGG pathways, and REVIGO was used to reduce the enrichment redundancy for a better visualization. The IPA was used to deduce the canonical pathways. More details in Section 3.

**Figure 2 cells-11-03496-f002:**
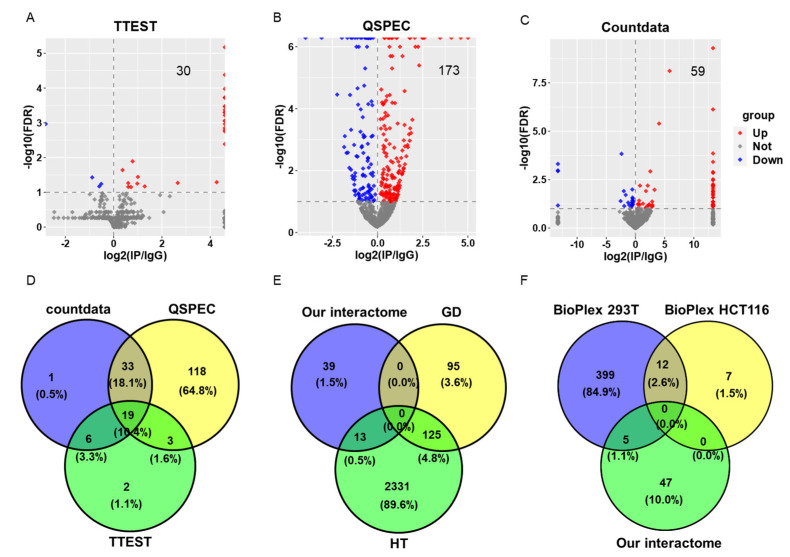
Proteins binding to 14-3-3 in the IP samples compared to the IgG samples. (**A**–**C**). Volcano plots of protein binding intensity in the IP samples compared to the IgG samples within three statistical methods, TTEST (**A**), Linux software, QSPEC (version: July 2021) (**B**), and R package, countdata (**C**). Proteins, with the cutoff −log10(FDR) > 1 and log2(IP/IgG) > 0, were regarded as proteins binding to 14-3-3 in mouse heart. (**D**). Venn diagrams of 14-3-3 interacting proteins obtained using the three statistical methods as indicated in the diagram. (**E**). Venn diagram showing the comparison of the 52 overlapped proteins and 14-3-3 interacting protein recorded in ANIA database, which contains two types of data: gold standard (GD) and high throughput data (HT). (**F**). Venn diagram showing the comparison of the 52 overlapped proteins and binding partners to 14-3-3 recorded in BioPlex database, which contains two types of protein–protein interactions: in HEK293T cell-line (BioPlex 293T) and in HCT116 cell-line (BioPlex HCT116).

**Figure 3 cells-11-03496-f003:**
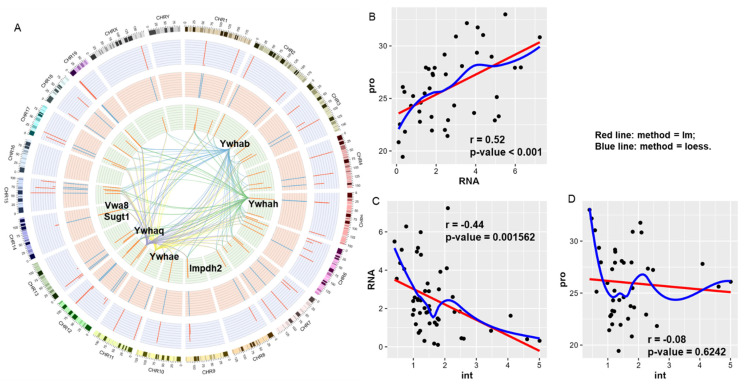
Correlation across transcriptome, proteome and 14-3-3 interactome. (**A**). Circos plot of transcriptome, proteome and 14-3-3 interactome. From the outer ring to the inner ring: chromosome, ideogram, public transcriptome, public proteome, and our 14-3-3 interactome. The central links from four 14-3-3 protein isoforms (Ywhab, Ywhah, Twhae and Ywhaq) to other proteins identified in our interactome delineate the potential bindings of 14-3-3 proteins to the targets. Three molecules, Impdh2, Vwa8 and Sugt1, represent the protein–protein interactions independent of transcription or translation. (**B**–**D**). Correlation scatter plots showing the correlation among the public transcriptome (RNA), proteome (pro) and 14-3-3 interactome (int). r, Pearson’s correlation coefficient.

**Figure 4 cells-11-03496-f004:**
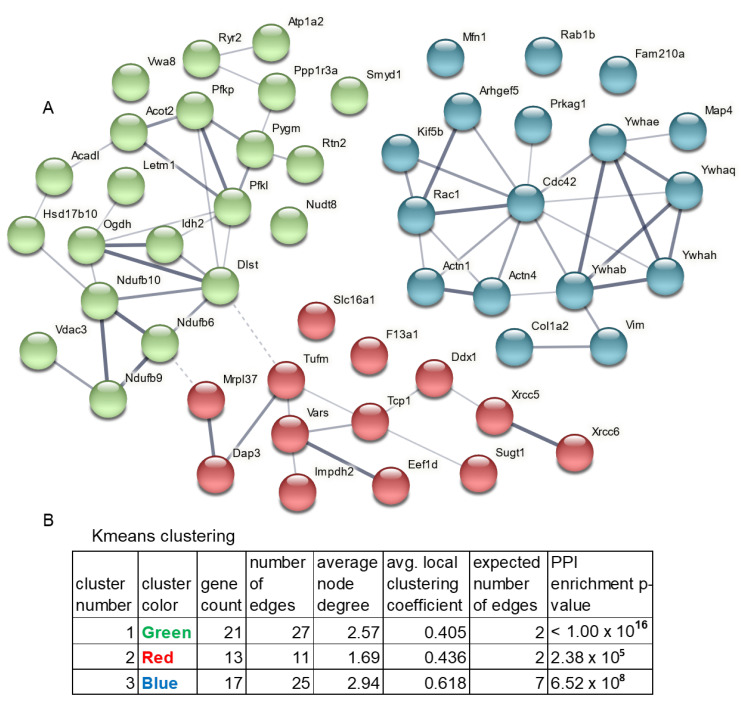
Protein−protein interaction network of the cardiac 14−3−3 interactome. (**A**). A global landscape of the cardiac 14−3−3 protein−protein interaction network. Nodes represent proteins. Edges represent interactions between nodes, and the line thickness indicates the strength of data support. Proteins clustered in the kmeans (*n* = 3) clustering method are labeled in the same color. (**B**). Sub−analysis of the three clusters and statistics.

**Figure 5 cells-11-03496-f005:**
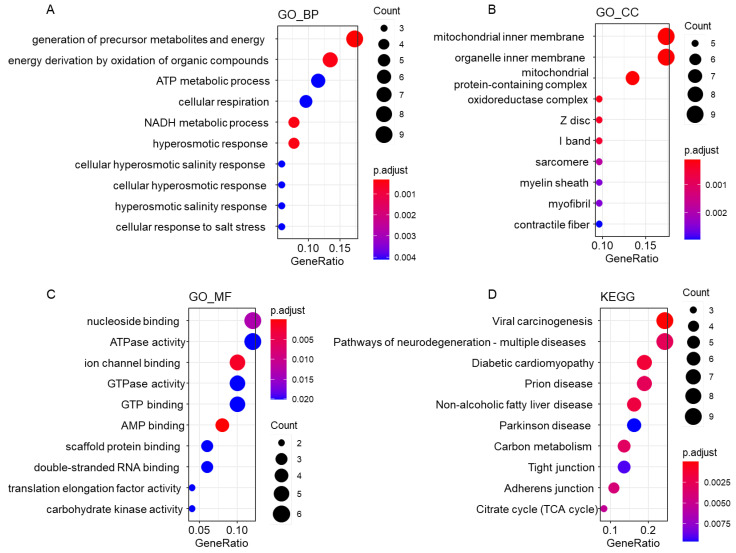
Functional enrichments from the 14-3-3 interactome using clusterProfiler. IDs of the 52 proteins were transferred into the ENTREZ gene IDs and imported into R package, clusterProfiler to conduct functional enrichment. Top ten terms with adjusted *p*-value < 0.05 were displayed in bubble charts. Four categories were GO_BP from Biological Process (Gene Ontology) database (**A**), GO_CC from Cellular Component (Gene Ontology) database (**B**), GO_MF from Molecular Function (Gene Ontology) database (**C**), and KEGG from KEGG pathways database (**D**).

**Figure 6 cells-11-03496-f006:**
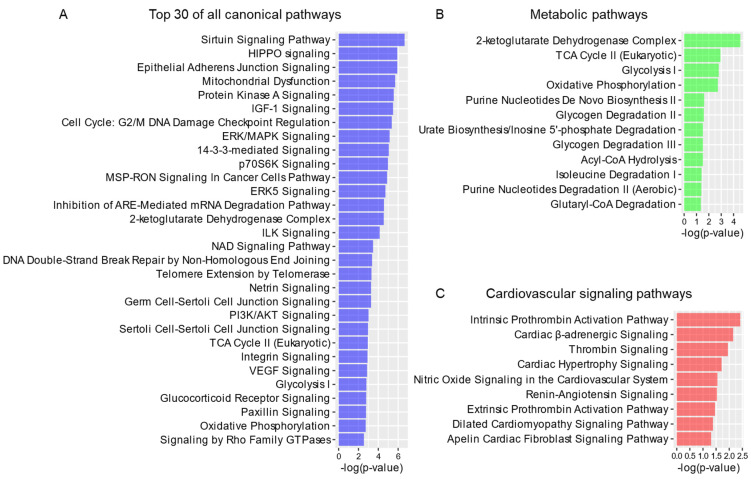
Ingenuity pathway analysis (IPA) of the cardiac 14−3−3 interactome. All the 52 proteins were imported into the Ingenuity pathway analysis (IPA) platform to conduct canonical pathways enrichment. Only pathways with −log10(*p*−value) > 1.3 (*p*−value < 0.05) were remained for subsequent analysis. (**A**) Top 30 canonical pathways sorted by −log10(*p*−value) from all canonical pathways, (**B**) all metabolic pathways and (**C**) all cardiovascular signaling pathways extracted from all canonical pathways were displayed in different colors.

**Figure 7 cells-11-03496-f007:**
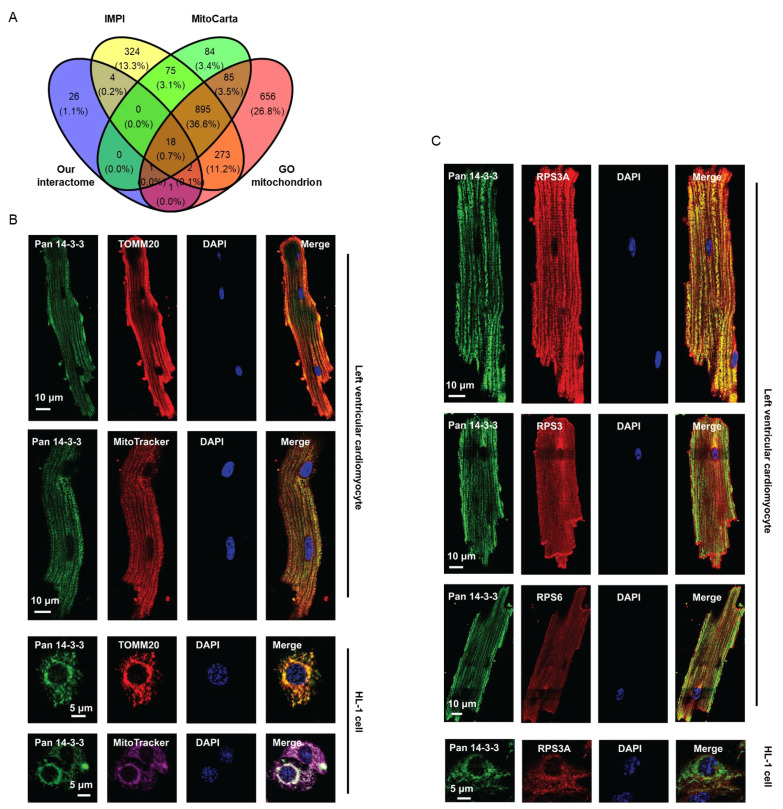
Representative localization of 14-3-3 interacting proteins in the mouse heart. (**A**). Venn diagram of proteins identified in our cardiac 14-3-3 interactome and the corresponding proteins within three mitochondrial protein databases, IMPI (Q2 2018), MitoCarta (2.0) and Gene Ontology (GO) mitochondrion. (**B**). Colocalization of 14-3-3 proteins and mitochondria in cardiomyocytes. Immunofluorescence images of indicated proteins and subcellular components and their merged images in HL-1 cells or primary left ventricular cardiomyocytes. Staining color: green, 14-3-3 proteins; red, TOMM20 proteins that localize in the outer membrane of mitochondria or MitoTracker indicating mitochondria; purple, MitoTracker in HL-1 cells; blue, nucleus; yellow or white, localization of 14-3-3 in mitochondria. Bars as shown in each row. (**C**). Colocalization of 14-3-3 proteins and ribosome in cardiomyocytes. Immunofluorescence images of indicated proteins and their merged images in HL-1 cells or primary left ventricular cardiomyocytes. Staining color: green, 14-3-3 proteins; red, three different ribosomal subunits, RPS6 (**C**) and RPS3, indicating ribosome; blue, nucleus; yellow, localization of 14-3-3 in ribosome. Bars as shown in each row.

**Figure 8 cells-11-03496-f008:**
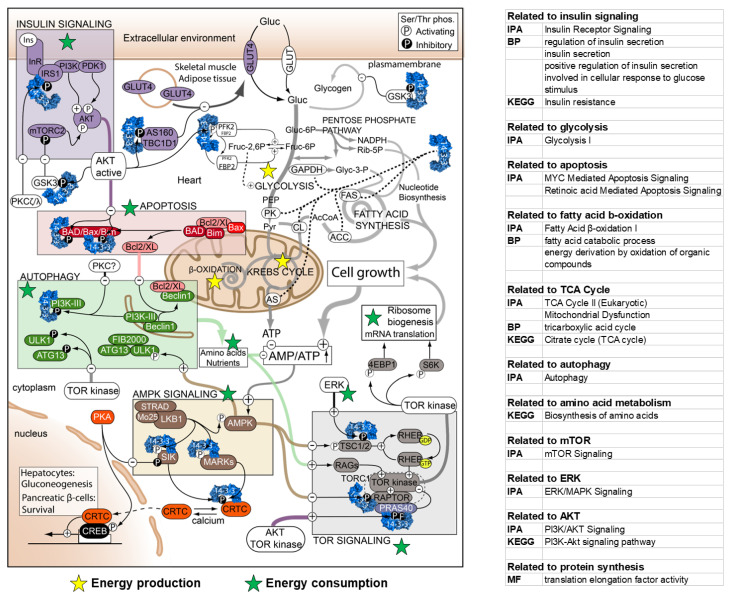
Involvement of 14-3-3 proteins in the regulation of cellular energy metabolism. The information acquired from the cardiac 14-3-3 interactome (Appendix A) is mapped to the cellular metabolism network summarized in a review of 14-3-3 interaction in noncardiac cells/tissues [7] (This figure was published in *Semin Cell Dev Biol*., 22(7), Kleppe R, Martinez A, Døskeland SO, Haavik J., The 14-3-3 proteins in regulation of cellular metabolism, 713-9, Copyright Elsevier) The integrated figure displays the involvement of 14-3-3 proteins in core metabolic pathways and regulatory signaling pathways in mouse heart, which consolidates and supplements the prior knowledge in noncardiac cells/tissues. In the original graph, 14-3-3 proteins are colored in blue and located close to their targets. Phosphorylation events are displayed as (P): stimulating phosphorylation on white background and inhibitory phosphorylation on black. We searched the results of related functional enrichments in clusterProfiler and IPA and then marked them in the graph with stars in different colors representing items related to energy production (yellow) or energy consumption (green): (1) energy production category contains proteins involved in glycolysis, Krebs (TCA) cycle, and β-oxidation; (2) energy consumption category contains proteins related to protein translation, and involved in insulin signaling, apoptosis, autophagy, amino acid biosynthesis, TOR signaling, AMPK signaling, and MAPK/ERK signaling.

**Figure 9 cells-11-03496-f009:**
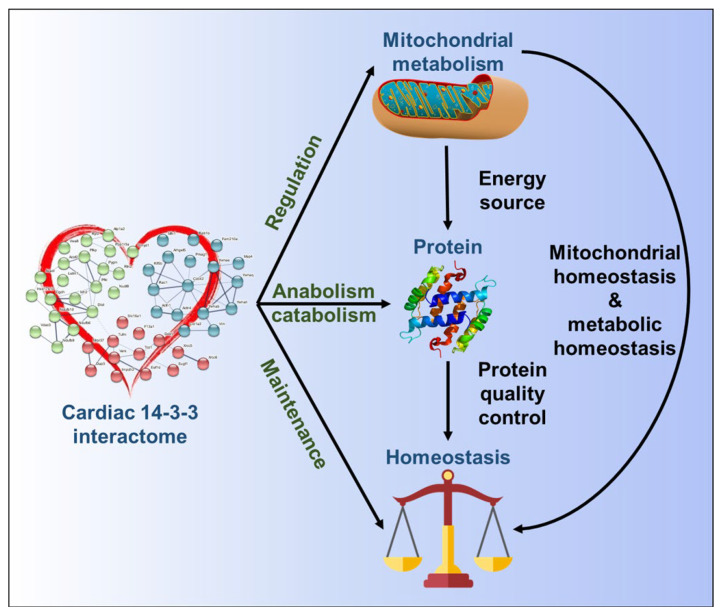
Schematic of the deduced mouse LV 14-3-3 protein–protein interactome. The 14-3-3 interactome binds to mitochondrial proteins and on this basis can regulate metabolism, thereby providing energy and resource supply and distribution, not only for heart contraction, but also for protein synthesis and degradation.

## Data Availability

The transcriptomic data of three-month old wildtype C57BL/6J male mouse hearts [39] (BioProject: PRJNA264807; SRA: SRP049245; Series: GSE62689; Samples: GSM1531478 (unstressed_ntg_1), GSM1531479 (unstressed_ntg_2), GSM1531480 (unstressed_ntg_3), GSM1531481 (unstressed_ntg_4)) were downloaded from NCBI GEO database. The proteomic data of 3-month old wildtype C57BL/6J male mouse hearts were downloaded from the Appendix A in a SILAC proteome project [40]. Our interactomic data were in Appendix A of this paper.

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
