# Peer review of "Proteomic Landscape and Deduced Functions of the Cardiac 14-3-3 Protein Interactome"

_cells, 2022, doi:10.3390/cells11213496_

Round 1
Reviewer 1 Report
Qu et al. set out to identify 14-3-3 interacting proteins in heart tissue. Although expanding the 14-3-3 interactome is an important effort, there are weaknesses in the premise and approach that undermine the conclusions and significance of this study. Also, numerous studies have focused on identifying novel 14-3-3 PPIs and it's not clear to this reviewer that this study adds to that body of literature.
1. A study like this, which focuses almost entirely on identifying 14-3-3 PPIs, must demonstrate a robust capture of the protein of interest. I don’t see convincing evidence that they got a robust IP of 14-3-3 form the LVs. PCA showed the IPs were different, but that doesn’t necessarily mean they had good IP of 14-3-3. They also saw 14-3-3 isoforms in MS data (which is not uncommon in any IP), but that still doesn't mean their IP was robust enough to see good capture of interacting partners. Furthermore, the coomassies of 14-3-3 and IgG IP look identical in panel A (they also really can’t be compared fairly if they’re run on separate gels). I would expect the authors could at least see a 14-3-3 band by Coomassie. If the level of 14-3-3 protein in the IP is below the level of detection by Coomassie, it calls into question the proteomics results. This might explain why their numbers of interactors are so low compared to previous approaches (figure 3).
2. What is the rationale for looking at 14-3-3 interacting proteins in heart tissue given that the 14-3-3 interactome has been studied in many other contexts? It'd be reasonable to expect that many of the most abundant 14-3-3-interacting proteins from other tissues would be present in their MS data, but that doesn't seem to be the case. It raises concern (together with point 1 above) that the bulk of this paper (which is mainly a statistical analysis of 14-3-3 IP MS data) is precariously focused on interactors that may not be true 14-3-3 interacting proteins. There is no validation of interactors, which would typically include IP immunoblots for specific candidates, and perhaps even mutation of the phospho-site that mediates 14-3-3 binding to confim that the interactor is a true phospho-dependent 14-3-3 interacting partner.
2. The ANIA database link listed in materials and methods isn’t working.
3. As mentioned above, numerous publications have used MS to elucidate the 14-3-3 interactome in various cells and tissues. Omitting these citations from the manuscript is a serious problem.
4. Fig 11 is a reproduction from another journal (PMID: 21888985). They do reference the paper (line 450) in the text but as far as I can tell it’s not clear that the image is a direct reproduction of a figure in that previously published review.
Reviewer 2 Report
The manuscript 'Structural Landscape and Deduced Functions of The Cardiac 14-3-3 Protein Interactome' by Qu et al. is one of the relatively few and valuable studies that analyzed the interactome of the widespread hub protein 14-3-3 in a specific cell type. The interactome analysis of 14-3-3 produced a number of novel and very interesting 14-3-3 partner proteins. The subsequent pathway and biological function analysis produced very useful insights in the possible role of 14-3-3 proteins in health and disease of the heart.
Rightfully, the authors claim in several places of the manuscript (lines 56-58, 839-841, 868-870, last sentence of the manuscript) that the regulatory network of 14-3-3 protein-protein interactions can be used for novel approaches in therapeutic intervention. It would add to the value of this manuscript, if the authors would discuss, how they envision this therapeutic intervention would look like (gene editing, mRNA, antibodies, small molecules?). There is some rich literature on small molecules, for example, that are being used to modulate 14-3-3 protein-protein interactions, but none of those are discussed in the paper. I believe the authors have slipped here a tremendous opportunity to help guiding the field to the concrete practical use of their finding for future therapies.
Reviewer 3 Report
Summary: In the article “ Structural Landscape and Deduced Functions of The Cardiac 2 14-3-3 Protein Interactome”, Qu et. al. attempt to determine which proteins bind to 14-3-3 proteins in the heart. After performing re-analysis of data from the literature, they perform semi-quantitative proteomics of immunoprecipitated 14-3-3 protein and its stable interaction partners, relative to capture with control antibody. They perform a number of bioinformatics analyses and provide a large number of figures that are very descriptive in nature. While accurately determining the 14-3-3 interactome in the heart would surely be a nice resource for the cardiometabolic research community, there is no central hypothesis for this work. With this in mind, I believe the text is too long and has several unnecessary figures—some should be moved to the online supplement and other not included at all. Further, key data analysis should be re-processed as described below to make the information a more reliable resource. I would recommend that a revised and more concise manuscript be considered for publication in Cells as a Brief Report (2 figures, 1 Table, 2500 words).
Primary Concerns:
1) Analysis of previous proteomics data: The data analyzed for Figure 1B is from a SILAC comparison of individual mice on the light (L) channel to a reference sample from SILAC-fed mice on the heavy (H) channel. Therefore, the L/H comparison for a given protein is its abundance relative to the amount of that same protein in the reference sample. Therefore, this information is only useful for comparison of abundances for a given protein in a mouse and the refence sample, and by proxy, to that same protein in all the other mice (as all mice use the same reference sample pool). Figure 1B, however, attempts to use the L/H ratios to compare across different proteins. This information is of no value. In the original paper by Geiger and co-workers, they do make estimates of abundance rank order between proteins, but these quantitative values are derived from the sum of each protein’s MS1 peak area in the light sample only, not from the H/L ratio. The authors of the current study should revisit this to produce a meaningful comparison of 14-3-3 protein abundances.
2) Quantitative analysis in current proteomics data: The authors collect data by Data Dependent Acquisition on a Q Exactive Orbitrap mass spectrometer. At a resolution of 70,000 for MS1 spectra, this raw data could very easily be re-analyzed with true label-free quantitation, where abundance values are derived from the peak areas of extracted ion chromatograms (XIC) produced from MS1 spectra (associated with peptides IDs from MS2 spectra). This type of analysis is now performed routinely and has been demonstrated to be more accurate than spectral counting, which is typically limited to low resolution mass spectrometers (e.g., linear ion trap) and is considered a semi-quantitative method. Further, quantitation using XICs is amenable to ranking different proteins by abundance (see point #1).
Minor Concerns:
3) Statistical comparisons: Re-processing the 14-3-3 interactome data with more modern methods (see point #2) should provide more reliable quantitative values that will likely produce much better p-values/adjusted p-values with a traditional t-test and BH correction. If the authors still favor comparing different statistical methods, I strongly suggest extending this to LIMMA, which has been used to great success in transcriptomics and is proving useful with modern quantitative proteomics data as well (Nucleic Acids Res. 43 (2015) e47.).
4) Unnecessary figures:
· Panel 3E is redundant, as this same information can be inferred from panel 3D.
· Table 1 shows information that all projects generate routinely and has almost no value to anyone other than the people doing the experiment. The manuscript will be much more interesting without this type of “space-filler” table in the main text. The authors should to select information that highlights what this study shows, not generic information that can be kept to their laboratory notebook. I would think a better us of a table would be something to do with the identification and annotation of the most confident 14-3-3 interaction partners.
· Are both Figure 6 and 7 needed in the main manuscript? They seem rather redundant, even though they are different analyses. The narrative of each analysis can remain in the text, but one of the figures (Figure 7 is nicer to look at in my opinion) could surely be in the supplement. Similarly, Figure 8 takes up 1.5 pages and is doesn’t really add anything beyond what is described about it in the text—it would work much better in the supplement.
5) Novel 14-3-3 binding partners. On line 312, the authors note that “39 proteins may be non-phosphorylated proteins”. This statement about phosphorylation status is presumably made because these are proteins not identified as 14-3-3 interactors previously. Is there also a possibility that these are phosphorylated proteins? The authors could investigate this in several ways: 1) The authors could search the already collected proteomics data with phosphorylation as a variable modification. 2) Even without identifying phosphorylation sites on these proteins, the authors could look for the presence of 14-3-3 binding motifs in the sequences surrounding phosphorylatable residues on the identified proteins (see phosphositeplus.org). Identifying a novel phosphoprotein interactor of 14-3-3 proteins could be a significant result leading to a new line of investigation (by this manuscript or future work).
6) Comparison with large-scale interactome efforts. The authors should consider cross-referencing their lists of 14-3-3 interactors with those present in BioPlex (tps://bioplex.hms.harvard.edu/), a large NIH-funded effort to map all protein-protein interactions in mammalian cells.
7) Comparison with proximity-based ligase approach. The authors could ideally express a fusion protein of a 14-3-3 protein and a proximity-based ligase, such as TurboID (Nat Biotechnol. 2018 Oct;36(9):880-887.), in cultured cells. These methods would be better suited for capturing dynamic protein-protein interactions that may be missed with immunoprecipitation. At the very least, the authors should reference these approaches as strategies for overcoming the limitations of affinity purification-MS would be appropriate.
Round 2
Reviewer 1 Report
Critiques were addressed reasonably
Author Response
Thank you very much.
Reviewer 3 Report
The manuscript has been significantly improved by addressing those concerns (from myself and the other reviewers) that could be resolved within the ten day resubmission deadline. My remaining three concerns, which can be easily resolved, are as follows: 1) In the title, the word "Structural" should be replaced with "Proteomic" as the current title seems to imply more biophysics-oriented techniques (e.g., x-ray, NMR, cryo-EM) were used to investigate the interactions. 2) The authors should define the "L" acronym as meaning " endogenous light (L)" on line 141. 3) I still feel that a full Research Article with eight figures is a lot for the experiments presented here, as the data seems more like a resource. My concern is that the article may be more widely read and cited if it was a more succinct Short Article. However, I defer to the editors as to whether the article format is appropriate.
Author Response
The manuscript has been significantly improved by addressing those concerns (from myself and the other reviewers) that could be resolved within the ten day resubmission deadline. My remaining three concerns, which can be easily resolved, are as follows:
1) In the title, the word "Structural" should be replaced with "Proteomic" as the current title seems to imply more biophysics-oriented techniques (e.g., x-ray, NMR, cryo-EM) were used to investigate the interactions.
Response:
Thank you for this good suggestion. We have revised the title to “Proteomic Landscape and Deduced Functions of The Cardiac 14-3-3 Protein Interactome”
2) The authors should define the "L" acronym as meaning " endogenous light (L)" on line 141.
Response:
Thank you for the professional advice. We have defined the term accordingly (Line 141).
3) I still feel that a full Research Article with eight figures is a lot for the experiments presented here, as the data seems more like a resource. My concern is that the article may be more widely read and cited if it was a more succinct Short Article. However, I defer to the editors as to whether the article format is appropriate.
Response:
Thank you very much. We appreciate your valuable suggestion. We will defer to the editors.